# A Taxonomic Revision of the Genus *Cionus* (Coleoptera, Curculionidae) from the Oriental Region

**DOI:** 10.3390/insects14070646

**Published:** 2023-07-18

**Authors:** Roberto Caldara, Michael Košťál

**Affiliations:** 1Via Lorenteggio, 37, 20146 Milano, Italy; 2Šoporňa 1602, SK-925 52 Šoporňa, Slovakia; michael.kostal@iol.cz

**Keywords:** Curculioninae, Cionini, *Cionus*, Oriental region, new species, new name, new synonymy

## Abstract

**Simple Summary:**

*Cionus* Clairville, 1798, is a genus of weevils distributed in Palaearctic, Afrotropical and Oriental regions. It belongs to the large subfamily Curculioninae of the family Curculionidae. This paper reviews all valid extant Oriental species of the genus based on morphological characters for the first time, describes one new species and gives a replacement name to another species due to primary homonymy. This is the third and final part of the revision of this genus after previous recent revisions of Palaearctic and Afrotropical species. Descriptions or redescriptions, illustrations of habitus and male genitalia, comparative biological notes, distribution, a detailed list of all examined specimens and a key to treated species are given.

**Abstract:**

Oriental species of the genus *Cionus* are herein revised for the first time. Eight species are recognized as distinct based on morphological characters of adults. One species is described as new: *C. ottomerkli* sp. nov., from India, whereas the name *vossi* (nom. nov.) is proposed for *Cionus flavoguttatus* Voss, 1957 (not Stierlin, 1893). The following new synonymy is established: *Cionus indicus* Desbrochers des Loges, 1890 (=*Cionus albosparsus* Faust, 1898 syn. nov.). Lectotypes of *Cionus albosparsus* Faust, 1898; *Cionus flavoguttatus* Voss, 1957; *Cionus indicus* Desbrochers des Loges, 1890; *Cionus obesus* Pascoe, 1883; and *Cionus tonkinensis* Wingelmüller, 1915, are designated.

## 1. Introduction

The genus *Cionus* Clairville, 1798, belongs to the tribe Cionini (Curculioninae, Curculionidae). Its distribution, as well as that of the whole tribe, includes Palaearctic, Afrotropical and Oriental regions, whereas it is absent, except for the accidental recent introduction of *Cionus scrophulariae* (Linnaeus, 1758), in Northeastern USA and Canada, as well as in the Nearctic, Neotropical and Australian regions. Currently, apart from *Cionus*, this tribe is composed of six other genera: *Cionellus* Reitter, 1904; *Cleopus* Dejean, 1821; *Nanomicrophyes* Pic, 1908; *Patialus* Pajni, Kumar and Rose, 1991; *Stereonychidius* Morimoto. 1962; and *Stereonychus* Suffrian. 1854. *Cionus* is the genus with the most species; it currently comprises about 120 valid species. In contrast, *Cleopus*, *Stereonychus* and *Nanomicrophyes* each includes less than ten species, and the other three genera include only one or two species. Recently, species of *Cionus* from the Palaearctic and Afrotropical regions were revised [1,2]. The following is the revision of the Oriental species, which have been very poorly studied.

## 2. Materials and Methods

### 2.1. Samples

About 250 specimens of Oriental *Cionus* were studied, including specimens of the type series of most taxa. Lectotypes were designated as appropriate according to Art. 74 and 75 of the International Code of Zoological Nomenclature [3], and all other specimens of the type series were labelled as paralectotypes. The rank of subspecific or infraspecific names was established according to Art. 45.5 and 45.6 of ICZN [3] and subsequent clarifications by Lingafelter and Nearns [4] and Alonso-Zarazaga et al. [5]. The unavailable names were noted as appropriate.

All data on labels are reported for each specimen examined in their exact sequence in specimens of the type series and ordered in a unified style in all other specimens.

### 2.2. Measurements

Measurements were made using an ocular micrometer in a Wild M8 stereoscopic microscope. Body length was measured from the anterior margin of eyes along the midline to the apex of elytra. The length of the rostrum (Rl) was measured in dorsal view, from the apex (excluding mandibles) to the anterior margin of eyes; the width (Rw) was measured at its widest point; and its relative length was expressed as ratios of length/width (Rl/Rw) and length of rostrum/length of pronotum (Rl/Pl). The length of the pronotum (Pl) was measured along the midline, from the apex to the base, whereas its width (Pw) was measured transversely at the widest point. The width of the pronotum was expressed as the ratio Pw/Pl. The length of elytra (El) was measured along the midline, from the transverse line joining the most anterior point of humeri to the apex, whereas its width (Ew) was measured transversely at the widest point. Proportions of elytra were expressed as the ratio El/Ew.

### 2.3. Description

The structure of descriptions follows one common model in order to make the terminology maximally unified by omitting common characters within the genus.

### 2.4. Diagnosis

A cluster of all characters determining a particular species was used.

### 2.5. Terminology

We followed the online glossary of weevil characters proposed in Lyal, C.H.C. (Ed.) Glossary of Weevil Characters. International Weevil Community Website. http://weevil.info/glossary-weevil-characters (accessed on 10 May 2023).

### 2.6. Bionomics

Regarding the systematics of the host plants, we followed APG IV [6].

### 2.7. Distribution

It is necessary to define here the Oriental region exactly since there are some discordances in its northern limits. In the recent Catalogue of Palaearctic Coleoptera [5] dealing with Curculionidae and in its previous volumes by Löbl and Smetana [7], parts of Bhutan, Nepal and India along the base of Himalaya were excluded from the Oriental region, thus not respecting the strict political boundaries of this last state. In contrast to Löbl and Smetana [7], knowing the Cionini of both Palaearctic and Oriental regions, we set our limits according to affinities among species; therefore, we decided to treat here species of the whole of India.

### 2.8. Illustrations

Photos of habitus were made with a high-resolution camera (Canon EOS 50D, Tokyo, Japan) and macro zoom lens (Canon MP-E 65 mm). Male genital structures were dissected and treated for five days in 10% KOH. Male genitalia were photographed in glycerol with the same camera under a laboratory microscope (Intraco Micro LMI T PC, Tachlovice, Czech Republic). The multilayer pictures were processed using the software Combine ZP. Female genitalia were not illustrated since they show only weak interspecific differences but often a remarkable intraspecific variability.

### 2.9. Acronyms and Abbreviations

Institutional depositories are abbreviated according to The Insect and Spider Collections of the World Website (http://hbs.bishopmuseum.org/codens/codens-inst.html, accessed on 20 May 2023). Abbreviations of host plant authors are reported only when mentioned for the first time as acronyms following the generally accepted list of botanist abbreviations from Wikipedia (https://en.wikipedia.org/wiki/List_of_botanists_by_author_abbreviation, accessed on 20 May 2023).

### 2.10. Depositories

Collections housing material studied in this revision are abbreviated as follows (with their curators in parentheses):
BMNHDepartment of Entomology, The Natural History Museum, London, U.K. (M. Barclay);DEIM Deutsches Entomologisches Institut, Müncheberg, Germany (L. Behne);FBCVcollection Friedhelm Bahr, Viersen, Germany;FTCMcollection Fabio Talamelli, S. Giovanni in Marignano, Italy;HNHM Hungarian Natural History Museum, Budapest, Hungary († O. Merkl);HWCB collection Herbert Winkelmann, Berlin, Germany;MKCScollection Michael Košťál, Šoporňa, Slovakia;MNHN Muséum National d’Histoire Naturelle, Paris, France (H. Perrin);MRAC Musée Royal de l’Afrique Centrale, Tervuren, Belgium (M. De Meyer);NHMB Naturhistorisches Museum, Basel, Switzerland (E. Sprecher);NHMW Naturhistorisches Museum, Wien (M. Jäch);NHRS Naturhistoriska Riksmuseet, Stockholm, Sweden (J. Bergsten);NMEGNaturkundemuseum, Erfurt, Germany (M. Hartmann);PBCS collection Piotr Białooki, Sopot, Poland;RCCM collection Roberto Caldara, Milano, Italy;SMTD Museum für Tierkunde, Dresden, Germany (O. Jäger, K.-D. Klass);ZISP Zoological Institute, Russian Academy of Sciences, St. Petersburg, Russia (B.A. Korotyaev).

### 2.11. Abbreviations

Eelytra;Ppronotum;Rrostrum;Sfunicular segment;Ttarsomere;Vventrite;Llength;Wwidth.

## 3. Results

### 3.1. Taxonomy

#### 3.1.1. *Cionus* Clairville

*Cionus* Clairville, 1798: 66 [8] [type species: *Curculio blattariae* Fabricius, 1792 (=*Curculio alauda* Herbst, 1784)]. Germar, 1821: 299 [9]. Schoenherr, 1838: 722 [10]; 1845: 178 [11]. Reitter, 1904: 49 [12]; 1912: 84 [13]; 1916: 232 [14]. Wingelmüller, 1914: 187 [15]; 1921: 102 [16]; 1937: 143 [17]. Hustache, 1932: 336 [18]. Hoffmann, 1958: 1211 [19]. Alonso-Zarazaga and Lyal, 1999: 76 [20]. Caldara and Korotyaev, 2002: 184 [21]. Caldara et al., 2014: 604 [22]. Košťál and Caldara, 2019: 7 [1]. Alonso-Zarazaga et al., 2023: 185 [5]. Caldara and Košťál, 2023: 7 [2].

*Mononyx* Brullé, 1839: 72 [23] (homonymy, not Laporte, 1832; type species *Mononyx variegatus* Brullé, 1839). Uyttenboogaart, 1937: 115 [24]. Košťál and Caldara, 2019: 7 [1]. Alonso-Zarazaga et al., 2023: 185 [5].

#### 3.1.2. Remarks

The genus *Cionus* was recently redescribed in two comprehensive revisions [1,2]. It is, however, important to present a brief diagnosis of the genus by indicating characteristics that are useful to separate species treated herein from species of other genera of the Cionini occurring in the Oriental region.

The genus *Cionus* can be distinguished from other genera of Cionini by one very distinctive apomorphy: a more or less sharply incised emargination on the anterior margin of prosternum. Moreover, the head between eyes is always narrower than the rostrum at base. Elytra of many Palaearctic species are characterized by one dorsal and often also one preapical black tomentous perisutural macula, which is absent in some Palaearctic, all Afrotropical (except *C. coniungens* Caldara and Košťál, 2023) and all Oriental (except *C. indicus* Desbrochers des Loges, 1890) species. The mesosternal process is flat, blunt at posterior margin; V1 is always longer than V2–5 combined, and V1–2 combined are always markedly (approximately 2.5× to 7.0×) longer than V3–4 combined. Male genitalia are always without parameroid lobes; the spermatheca is similar in the shape among species, with robust body and long, thin, strongly curved cornu. Finally, unlike other genera of the Cionini, most species of *Cionus* have asymmetrical claws, with the outer claw being longer than the inner claw in males, especially on the protarsi.

In addition to the distinctive prosternal emargination, the genus *Cionus* differs from the following other genera of the Cionini present in the Oriental region by following character possession of two claws vs. only one claw in *Stereonychus* and *Stereonychidius*; asymmetrical claws in males of most species; more or less convex body outline in lateral view vs. *Cleopus* having body moderately flat; lack of a concave mesosternal process; and by a distinct fovea in anterior 2/3 of median portion of the metasternum vs. indistinct fovea in *Patialus*.

#### 3.1.3. List of Oriental Species


**1. *Cionus indicus* Desbrochers des Loges, 1890**


= *Cionus albosparsus* Faust, 1898 **(syn. nov.)**


**2. Cionus radermacherae Voss, 1934**



**3. Cionus vossi nom. nov.**


= *Cionus flavoguttatus* Voss, 1957 (not Stierlin, 1893)


**4. *Cionus meleagris* Marshall, 1926**



**5. Cionus ottomerkli sp. nov.**



**6. *Cionus albopunctatus* Aurivillius, 1892**



**7. *Cionus tonkinensis* Wingelmüller, 1915**



**8. *Cionus obesus* Pascoe, 1883**


#### 3.1.4. Treatment of Species

**1.** ***Cionus indicus* Desbrochers des Loges** (Figure 1a–f)

*Cionus indicus* Desbrochers des Loges, 1890: 216 [25]. Marshall, 1926: 367 [26].

*Cionus albosparsus* Faust, 1898: 305 [27]. Marshall, 1926: 367 [26]. (**syn. nov.**).

**Type locality.** Dam-Dim (West Bengal, Northeastern India).

**Type series.** According to the original description, *Cionus indicus* was described from specimens collected at Dam-Dim (Northeastern India) preserved in Desbrochers des Loges’ collection. In MNHN, we examined one damaged female specimen lacking most scales of the dorsal and ventral vestiture, pinned in the right elytron and labelled “n. sp. [two illegible words]/Calcutta/Dam-Dim/Collection Desbrochers/Museum Paris, coll. A. Clerc/coll. Paris A. Clerc” (lectotype here designated).

**Synonyms.** *Cionus albosparsus* was described from Kanara (Karnataka, Western–Central India). We found the following three female syntypes labelled as follows: “[small quadrate golden paper]/Kanara, Andrewes/Coll. J. Faust Ankauf 1900/Type” (SMTD, lectotype here designated); “[small quadrate goldish paper]/Kanara, Andrewes/albosparsus Fst./Coll. J. Faust Ankauf 1900/Type” (SMTD, paralectotype); and “Para-type [white round card with yellow edges]/Kanara/G.A.K. Coll., B.M. 1950−255/Cionus albosparsus n.sp.” (BMNH, paralectotype). As supposed by Marshall (1926) based on the original description, “*C. indicus* is only a damaged specimen of *C. albosparsus* lacking most scales”.

**Diagnosis.** Pronotum between midline and sides with two small tufts of erect white scales. Elytra with rounded, large black macula surrounded by whitish scales laterally reaching stria 2, which is slightly sinuate around the macula; all striae well visible. Claws asymmetrical in male.

**Redescription.** Male.

Body. Stout, globose.

Head. Rostrum moderately slender, medium long (Rl/Rw 4.27–4.33, Rl/Pl 1.27–1.34), black, in lateral view distinctly curved, weakly tapered from antennal insertion to apex; in dorsal view parallel-sided, with intermixed whitish and light brown, moderately dense, subelliptical (l/w 5–8), recumbent to subrecumbent scales to near apex. Head between eyes narrow, 1/4 as wide as rostrum at base. Eyes flat. Antennae dark brown, inserted just behind midlength, funicle distinctly shorter than scape, S1 more robust, 1.3× longer than S2, twice as long as wide, S2 1.3× as long as wide, S3–5 distinctly transverse; club elongate oval, regularly pubescent, distinctly longer than funicle.

Thorax. Pronotum black, with dense, moderately deep punctures, intervals between punctures narrow, smooth, shining, moderately visible between recumbent, moderately dense, elliptical (l/w 5–7) intermixed whitish and light brown scales, all directed forwards, between midline and sides with two small tufts of erect white scales; conical, transverse (Pw/Pl 1.70–1.75), widest at base, sides slightly rounded, moderately convex, without tubercles. Prosternum with anterior margin with deep emargination almost reaching coxae, margins of emargination flat, without canal. Scutellar shield subtriangular, with moderately dense intermixed whitish and light brown scales.

Elytra. Black, globose (El/Ew 1.01–1.04), widest at middle, at base distinctly wider than pronotum (Ew/Pw 1.80–1.86), moderately rounded, disc distinctly convex; interstriae slightly convex, with shallow, dense irregular punctures; scales mainly elliptical (l/w 5–7), recumbent, moderately dense, light brown, a few scales white, sparse, slightly broader than brown ones; before middle with large rounded macula of black scales covering interstriae 1 and 2 surrounded by whitish scales forming additional small rectangular spots on interstria 1 behind dark spot, in apical third and very small spot on interstria 5 between basal and middle third; striae moderately visible, with deep punctures, intervals between punctures narrow, at same level as interstriae, stria 2 only slightly sinuate around dark spot.

Venter. Mesosternal process slightly convex, large, short, reaching only base of coxae, moderately emarginate. Mesosternum as long as V1. Abdomen with irregular, dense, moderately deep punctures moderately visible between dense, elongate intermixed whitish and light brown, scales, which are more elongate, seta-like at middle; V1 1.67–1.70× as long as V2; V1–2 3.40–3.47× as long as V3–4, V3–4 1.35–1.38× as long as V5.

Legs. Femora black with intermixed whitish and light brown elliptical (l/w 5–7), recumbent, moderately dense scales, with moderately stout sharp teeth; tibiae black, without uncus; tarsi black, T1 1.3× as long as wide, T2 transverse, T3 distinctly bilobed, distinctly broader than T2, onychium as long as T1–3 combined; one claw is 2/3 as long as the other.

Penis. Figure 1d–f, body parallel-sided with obtusely pointed tip and poorly sclerotized flagellum.

Female. Rostrum slightly longer (Rl/Rw 4.90–4.98; Rl/Pl 1.37–1.45), claws symmetrical.

**Variability.** Length 4.0–4.9 mm. Sometimes the small white spot on interstria 5 is absent.

**Comparative notes.** This species is easily distinguishable from the other Oriental taxa of the genus by a spot of dark scales on the disk of elytra, similar to that in most Palaearctic species. In contrast, *C. indicus* lacks the apical spot.

**Biology.** One specimen was collected in Myanmar on *Premna pyramidata* Wall. ex Schauer (Lamiaceae). This finding of a single specimen is almost certainly an accidental occurrence.

**Distribution.** India, Bangladesh, Myanmar, Laos, Thailand, Vietnam.

**Non-type specimens examined.** India*:* Assam (1, NMEG); Assam, Nameri National Park, Tezpur 60 km N, legg. Afonin and Siniaev (6, FTCM); Jashpur district, Chainpur env., Barway Mission (6, MRAC); Kanara (1, BMNH); Malabar (1, BMNH; 4, MNHN); Mombay, Mathéran (1, BMNH; 1, SMTD); Nilgiri Hills, leg. Andrewes (1, BMNH); Pegu (2, MNHN); Walayar, IX.1952, leg. Subramaniam (1, BMNH). Bangladesh: Silhet, Chandkhiva, leg. Sherwill (1, BMNH). Myanmar: Mandalay, Yeni Res., Pyinmana, 7.VI.1934, on foliage *Premna pyramidata*, leg. Desai (1, BMNH); Tharrawaddy (1, BMNH). Laos: Ban Vay (3, NHMW). Thailand: Nan, Bo Khua, 1.–11.V, leg. Moravec (1, PBCS). Vietnam: Bao-lac, III.1902 (1, BMNH; 1, MNHN).

**2.** ***Cionus radermacherae* Voss** (Figure 2a–f)

*Cionus radermacherae* Voss, 1934: 272 [28]. Kalshoven, 1956: 85 [29].

**Type locality.** Tjiamis (Mount Sawel, Java, Indonesia).

**Type series.** This species was described from specimens collected in Tjiamis (Java). We examined the holotype labelled “Java, Ges. Sawel, 500 m, Tjiamis, 21.III.1933, L. G. E Kalshoven/Radermachera gigantea/Type/Holotypus/Cionus radermacherae n. sp. E. Voẞ det.” (DEIM).

**Diagnosis.** Rostrum long and thin, especially in female. Pronotum between midline and sides with two little spots of suberect white scales simulating small tubercles. Even elytral interstriae covered with whitish grey scales. Claws asymmetrical in both sexes.

**Redescription.** Male.

Body. Stout, globose.

Head. Rostrum relatively thin, moderately long (Rl/Rw 4.70–4.74, Rl/Pl 1.16–1.21), black, in lateral view curved, slightly tapered from antennal insertion to apex; in dorsal view almost parallel-sided from base to apex, tricarinate; with recumbent, sparse, subtle, short (l/w 2–4) light brown scales in basal half. Head between eyes very narrow, 1/4 as wide as rostrum at base. Eyes flat. Antennae brown, inserted behind midlength, scape long, funicle slightly shorter than scape, S1 moderately more robust than and as long as S2, 1.7× as long as wide, S2 twice as long as wide, S3–5 transverse, club elongate oval, regularly pubescent, slightly shorter than funicle.

Thorax. Pronotum black, without tubercles, punctures poorly visible, moderately deep, dense, intervals between punctures shining, finely rugose, hardly visible between recumbent, elliptical (l/w 4–7) whitish and brown scales, lighter ones sparse, intermixed to darker ones, partly subrecumbent forming two small spots between middle and sides, and additional indistinct spots; conical, transverse (Pw/Pl 1.73–1.75), widest at base, sides in basal 2/3 very slightly rounded, moderately convex. Anterior margin of prosternum with deep emargination, without canal. Scutellar shield cordiform, with dense brown scales.

Elytra. Black, globose (El/Ew 1.08–1.11), widest at middle (Ew/Pw 1.70–1.74), humeri prominent, rounded, sides very slightly rounded, disc distinctly convex; odd interstriae weakly convex, even interstriae flat, with small shallow sparse punctures, intervals smooth, shining; scales recumbent, on even interstriae whitish, on odd interstriae brown and whitish, brown scales forming large rectangular (l/w 3–6) maculae, white scales forming small subquadrate spots; striae slightly sinuate, with almost regular large deep punctures as wide as interstriae, intervals between punctures narrow, situated lower than interstriae.

Venter. Mesosternal process is slightly convex, moderately emarginate. Mesosternum as long as V1. Abdomen with dense, irregularly arranged, moderately deep punctures feebly visible between moderately dense, elongate (more elongate, hair-like along midline and on tuft on V2); V1 1.90–1.95× as long as V2; V1–2 4.77–4.80× as long as V3–4, V3–4 0.90–0.94× as long as V5.

Legs. Femora black with more or less dense whitish scales, with stout sharp teeth; tibiae black, without uncus; tarsi black, T1 1.2× as long as wide, T2 transverse, T3 distinctly bilobed, distinctly broader than T2, onychium slightly shorter than T1–3 combined; claws asymmetrical, with one claw being half as long as the other.

Penis. Figure 2d–f, body with subparallel sides, poorly sclerotized long flagellum, and two convergent, symmetrical long sclerites in its apical half.

Female. Rostrum distinctly longer (Rl/Rw 5.50–5.57, Rl/Pl 1.30–1.34), considerably arcuate, scales on abdomen elongated, thin, only slightly longer in middle than at sides, not hair-like, claws slightly asymmetrical.

**Variability.** Length 3.7–4.7 mm. There are no noteworthy differences among the few specimens examined.

**Comparative notes.** Due to the length of the rostrum, especially in females, this species can be confused only with *C. vossi* from which it differs by the dorsal pattern similar to that of *C. albopunctatus*.

**Biology.** Collected on *Radermachera gigantea* (Blume) Miq. (Bignoniaceae), where the adults were observed “on the terminal shoots of saplings and on suckers from stumps of newly felled trees” [29].

**Distribution.** Myanmar, Malaysia, Indonesia.

**Non-type material examined.** Myanmar*:* Tenasserim, Javoy, leg Doherty (1, BMNH). Malaysia: Kedah, Jitra, Catciment Area, 9–14.IV.1928 (1, BMNH). Indonesia: M.-Java, Sarangan, Lawoe-Geb. 1500–2000 m, 1927-28, leg. Overbeck (1, SMTD); W Sumatra prov. Kerinci Seblat N. P., 24 km NE Tapan, Muara Sako E env. 2°05′ S 101°15′ E, 400–550 m, 4–18.III.2003, leg. Dembicky (1, NHMB).

**3.** ***Cionus vossi* nom. nov.** (Figure 3a,b)

LSID urn:lsid:zoobank.org:act:8F3BC927-FAA9-46F5-8BCC-C1357714A885

*Cionus flavoguttatus* Voss, 1957: 111 [30] (not Stierlin, 1893)

**Type locality.** Inginiyagala (Uva Province, Sri Lanka).

**Type series.** Voss [29] described *Cionus flavoguttatus* from a single specimen collected in Inginiyagala (Sri Lanka), which we examined in NHMB. It is a female labelled “TYPUS/CEYLON Uva, Inginiyagala, 2.IX.53, F. Keiser/Cionus flavoguttatus n. sp. E. Voss det., 1956”.

According to the Code [3] (Art. 57.2, 57.4) Voss’ name cannot be used as valid because of primary homonymy with *Cionus* (*Stereonychus*) *fraxini* var. *flavoguttatus* Stierlin, 1893. The latter name is here considered available (i.e., subspecific) according to the Art. 45.5 and 45.6 because not used as “unambiguously infrasubspecific” by Stierlin [31] (see also Materials and Methods). Therefore, in the absence of synonyms (Art. 60.3) we propose the name *Cionus vossi*
**nom. nov.** for *C. flavoguttatus* Voss, as an objective replacement by using the same type specimen (Art. 72.7).

**Diagnosis.** Rostrum medium thin, elongate. Pronotum without tubercles, with two small spots of recumbent white scales between the midline and sides. Even elytral interstriae covered with brown scales, interstria 1 at midlength with two rectangular spots of white scales.

**Redescription.** Female.

Body. Stout, globose.

Head. Rostrum medium thin, elongate (Rl/Rw 4.72–4.75, Rl/Pl 1.18–1.22), black, in lateral view moderately curved, slightly tapered from antennal insertion to apex; in dorsal view from base to apex almost parallel-sided; in basal half with recumbent, sparse, subtle, short (l/w 2–4) light brown scales. Head between eyes very narrow, 1/4 as wide as rostrum at base. Eyes flat. Antennae brown, inserted at midlength, scape long, funicle slightly shorter than scape, S1 moderately more robust than and as long as S2, 1.7× as long as wide, S2 twice as long as wide, S3–5 transverse, club elongate oval, regularly pubescent, slightly shorter than funicle.

Thorax. Pronotum black, without tubercles, punctures feebly visible, moderately deep, dense, intervals between punctures shining, finely rugose, hardly visible between recumbent, elliptical (l/w 4–7) whitish and brown scales, lighter ones sparse, intermixed to darker ones forming two small spots on each side between middle and sides; conical, transverse (Pw/Pl 1.71–1.75), widest at base, sides in basal 2/3 moderately rounded, moderately convex. Anterior margin of prosternum with deep emargination, without canal. Scutellar shield cordiform, with dense brown scales.

Elytra. Black, globose (El/Ew 1.08–1.12), widest at middle (Ew/Pw 1.70–1.73), humeri prominent, rounded, sides very slightly rounded, disc distinctly convex; interstriae weakly convex, with small shallow sparse punctures, intervals smooth, shining; scales recumbent, brown and whitish, latter ones elliptical forming small subquadrate spots on odd interstriae and two slightly larger rectangular spots on interstria 1 behind scutellum and at middle (l/w 4–7); striae slightly sinuate, with almost regular, large deep punctures as wide as interstriae, intervals between punctures narrow, situated at level of interstriae.

Venter. Mesosternal process is slightly convex, moderately emarginate. Mesosternum as long as V1. Abdomen with dense, irregularly arranged, moderately deep punctures, feebly visible between moderately dense, rectangular, elongate but not hair-like, whitish yellow scales; V1 1.75× as long as V2; V1–2 4.01–4.05× as long as V3–4, V3–4 0.83–0.86× as long as V5.

Legs. Brown, femora with intermixed dark brown and light brown scales, with stout sharp teeth; tibiae without uncus; tarsi with T1 1.2× as long as wide, T2 transverse, T3 distinctly bilobed, distinctly broader than T2, onychium slightly shorter than segments 1–3 combined; claws symmetrical.

Male. Unknown

**Variability.** Length 3.7–4.7 mm. There are noteworthy differences among four specimens examined.

**Etymology.** We dedicate this species to Eduard Voss (1884–1974), one of the most eminent experts in Curculionoidea of the 20th century.

**Comparative notes.** This species as well as *C. radermacherae* is easily distinguishable from all other Oriental species bearing similar small spots of whitish scales on pronotum and elytra by the long and thin rostrum in females.

**Distribution.** Central and Southern India, Sri Lanka.

**Non-type material examined.** India: Ayur, North Salem, F. R. I. Sandal Insect Survey, 11.IX.1930 (1, BMNH); Madhya Pradesh, Dekan (1, SMTD); S. India, Tamil Nadu, Nilgiri Hills, 11 km SE Kotagiri, 1100 ± 100 m, 11°24′ N 76°56′ E, Kunchappanal, 3–15.V.2002, leg. Pacholátko (1, NHMB).

**4.** ***Cionus meleagris* Marshall** (Figure 4a–f)

*Cionus meleagris* Marshall, 1926: 363, 368 [26].

**Type locality.** Nilgiri Hills (Kerala, Southwestern India).

**Type series.** This species was described from five somewhat damaged specimens collected in Nilgiri Hills. In BMNH, we examined all of these specimens labelled as follows: “Type [round white card with red margins]/Nilgiri Hills, A.K. Weld Downing/Pres. by Imp. Bur. Ent. Brit. Mus. 1926-95 [upturned card]/Cionus meleagris TYPE: ♂ Mshl.” (male, lectotype here designated); “Co-, type [round white card with yellow margins]/Nilgiri Hills, A.K. Weld Downing/Pres. by Imp. Bur. Ent. Brit. Mus. 1926-95 [upturned card]/Cionus meleagris COTYPE: ♀ Mshl.” (female, paralectotype); 3 exx. “Para-, type [round white card with yellow margins]/Nilgiri Hills, A.K. Weld Downing/G.A.K. Marshall Coll., B.M. 1950-255 [upturned card]/Cionus meleagris COTYPE: ♂ Mshl.” (three male specimens, paralectotypes).

**Diagnosis.** Rostrum stout, short in both sexes. Pronotum without tubercles, with small spots of light scales both on disc and at sides. Elytra with many light spots, with distinctly prominent, right-angled humeri. V2 in male with tuft of hair-like scales. Claws asymmetrical in male.

**Redescription.** Male.

Body. Stout, globose.

Head. Rostrum stout, short (Rl/Rw 2.95–3.00, Rl/Pl 1.26–1.29), black, in lateral view weakly curved, slightly tapered from antennal insertion to apex; in dorsal view moderately enlarged from base to apex; with five distinct carinae (one median and two lateral at each side), with recumbent, moderately dense, moderately elongate (l/w 5–7) whitish and light brown scales near to apex. Head between eyes very narrow, 1/4 as wide as rostrum at base. Eyes flat. Antennae brown, inserted between middle and apical 1/3, scape long, funicle shorter than scape, S1 moderately more robust than and as long as S2, 1.8× as long as wide, S2 twice as long as wide, S3–5 transverse, club elongate oval, regularly pubescent, slightly shorter than funicle.

Thorax. Pronotum black, without protuberances, punctures feebly visible, moderately deep, dense, intervals between punctures shining, finely rugose; scales recumbent, elliptical (l/w 4–7) whitish and light brown, former ones forming several confluent small spots; conical, transverse (Pw/Pl 1.68–1.72), widest at base, sides subrectilinear, moderately convex. Anterior margin of prosternum with deep emargination, without canal. Scutellar shield cordiform, with dense light brown scales.

Elytra. Black, globose (El/Ew 1.02–1.04), widest at middle (Ew/Pw 1.42–1.46), at base distinctly wider than pronotum, humeri distinctly prominent, right-angled, sides very slightly rounded, disc distinctly convex; interstriae weakly convex (even ones very slightly more than odd ones), with small, shallow sparse punctures, intervals smooth, shining; even interstriae with recumbent to subrecumbent, elliptical (l/w 4–7) greyish and light brown intermixed scales, odd interstriae with whitish and dark brown scales forming small subquadrate spots; striae straight, with ill-visible deep punctures, as wide as half of interstriae, intervals between punctures narrow, situated at level of interstriae.

Venter. Mesosternal process is slightly convex, slightly emarginate. Mesosternum almost as long as V1. Abdomen with dense, irregular, moderately deep punctures feebly visible between elongate whitish scales, more elongate, hair-like in middle especially of V1 and V2, suberect denser scales forming distinct tuft on apical part of V2; V1 1.55–1.60× as long as V2; V1–2 5.10× as long as V3–4, V3–4 0.80–0.85× as long as V5.

Legs. Femora black with intermixed dark brown and light brown scales, with stout sharp teeth; tibiae black, without uncus; tarsi brown, T1 1.3× as long as wide, T2 transverse, T3 distinctly bilobed, distinctly broader than T2, onychium as long as T1–3 combined; claws are asymmetrical, with one claw being 2/3 as long as the other.

Penis. Figure 4d–f, body with slightly convergent sides, blunted tip, very long thin flagellum bifurcated at base, and with two convergent symmetrical sclerites in its apical half.

Female. Rostrum slightly longer (Rl/Rw 3.27–3.30; Rl/Pl 1.24–1.27), scales on abdomen only slightly longer in middle than at sides, not hair-like, claws almost symmetrical.

**Variability.** Length 4.0–4.3 mm. There are no noteworthy differences among the specimens of the type series.

**Comparative notes.** This species differs from *C. ottomerkli* by the shorter rostrum, which is not tapered from the antennal insertion to the apex, the pronotum covered with small spots of light scales both on the disc and at sides, the elytral interstriae 8 and 9 only slightly sinuate.

**Distribution.** Southern India (Tamil Nadu).

Non-type material examined. No specimens.

**5.** ***Cionus ottomerkli* sp. nov.** (Figure 5a–f)

LSID urn:lsid:zoobank.org:act:073134D2-86E3-484B-8C92-2E0941429C5A

**Type locality.** Ramdurg (Karnataka State, Southern India).

**Type series.** Holotype “India or. Ramandorog [=Ramdurg], [leg.] Katona [nickname of Kálmán Kittenberger] 1919” (male, HNHM). Paratypes. Same data as holotype (8, HNGM, MKCS, RCCM).

**Diagnosis.** Integument reddish brown. Rostrum stout, short. Pronotum without tubercles, densely covered with light scales at sides. Elytra with many light spots, with prominent rounded humeri. V2 with tuft of hair-like scales. Claws asymmetrical in male.

**Description.** Male.

Body. Stout, globose. Integument completely reddish brown.

Head. Rostrum stout, short (Rl/Rw 3.30–3.34, Rl/Pl 1.30–1.35), in lateral view moderately curved, very slightly tapered from antennal insertion to apex; in dorsal view of same width from base to apex; with distinct carina along midline, with recumbent, moderately dense, elongate (l/w 4–7) whitish and light brown scales near to apex. Head between eyes very narrow, 1/4 as wide as rostrum at base. Eyes flat. Antennae inserted between middle and apical 1/3, scape long, funicle shorter than scape, S1 slightly more robust than and as long as S2, twice as long as wide, S2 twice as long as wide, S3–5 transverse, club elongate oval, regularly pubescent, slightly shorter than funicle.

Thorax. Pronotum without tubercles, punctures feebly visible, moderately deep, dense, intervals between punctures shining, finely rugose; scales elliptical (l/w 4–7), recumbent whitish and light brown, former ones forming several small spots; conical, transverse (Pw/Pl 1.70–1.73), widest at base, sides subrectilinear, moderately convex. Anterior margin of prosternum with deep emargination, without canal. Scutellar shield cordiform, with dense light brown scales.

Elytra. Globose (El/Ew 1.02–1.05), widest at middle (Ew/Pw 1.42–1.46), humeri prominent, rounded, sides very slightly rounded, disc distinctly convex; interstriae weakly convex (even ones very slightly more convex than odd ones), with small, shallow sparse punctures, intervals smooth, shining; even interstriae with recumbent to subrecumbent, elliptical (l/w 4–7), intermixed greyish and light brown scales, whitish and dark brown scales on odd interstriae forming small subquadrate spots; striae 4–9 sinuate, at basal third especially striae 8 and 9 angulate, with moderately visible deep punctures, as wide as half of interstriae, intervals between punctures narrow, at same level as interstriae.

Venter. Mesosternal process is slightly convex, slightly emarginate. Mesosternum as long as V1. Abdomen with dense, irregular, moderately deep punctures, moderately visible between elongate whitish scales, more elongate, hair-like in middle especially of V1 and V2, suberect, denser scales forming tuft in apical part of V2; V1 1.50–1.56× as long as V2; V1–2 5.10–5.15× as long as V3–4, 3–4 0.78–0.85× as long as V5.

Legs. Femora with intermixed dark brown and light brown scales, with stout sharp teeth; tibiae without uncus; tarsi with T1 1.3× as long as wide, T2 transverse, T3 distinctly bilobed and distinctly broader than tarsomere T2, onychium as long as T1–3 combined; claws are asymmetrical, with one claw being 2/3 as long as the other.

Penis. Figure 5d–f, body with subparallel sides, blunted tip, and long, thin, poorly sclerotized flagellum.

Female. rostrum longer (Rl/Rw 4.02–4.08; Rl/Pl 1.35–1.38), scales on abdomen only slightly longer in middle than at sides, claws symmetrical.

**Variability.** Length 4.2–4.5 mm. There are no noteworthy differences among specimens of the type series.

**Etymology.** This species is named in memory of our friend and colleague Ottó Merkl (1957–2021), prematurely passed away, who kindly searched and sent us several specimens preserved in the collections (HNHM), which were very important for our study.

**Comparative notes.** This species is closely related to *C. meleagris*, from which it differs by the moderately longer rostrum being in lateral view tapered from the antennal insertion to the apex, the pronotum densely covered with light scales at its sides, distinctly sinuate elytral interstriae 8 and 9 in their basal third.

**Distribution.** Southern India (Tamil Nadu).

**Non-type material examined.** No other specimens apart from those of the type series.

**6.** ***Cionus albopunctatus* Aurivillius** (Figure 6a–f)

*Cionus albopunctatus* Aurivillius, 1892: 218 [32]. Faust, 1894: 239 [33]. Marshall, 1926: 368 [26].

**Type locality.** Luang Prabang (Laos).

**Type series.** *Cionus albopunctatus* Aurivillius was described from specimens collected in Luang Prabang (Laos). One syntype was examined in NHRS by the first author.

**Diagnosis.** Pronotum distinctly conical, with rectilinear sides, two small tubercles, and many small spots of light scales of which two cover tubercles. Elytra with many small light spots, interstria 3 at base moderately carinate. Claws in male slightly asymmetrical.

**Redescription.** Male.

Body. Stout, globose.

Head. Rostrum stout, short (Rl/Rw 3.20–3.28, Rl/Pl 1.20–1.25), black, in lateral view moderately curved, slightly tapered from antennal insertion to apex; in dorsal view slightly enlarged from base to apex; with recumbent, sparse, subtle (l/w 3–6) whitish and light brown scales near to apex. Head between eyes very narrow, 1/4 as wide as rostrum at base. Eyes flat. Antennae dark brown, inserted between middle and apical 1/3, scape long, funicle shorter than scape, S1 slightly more robust than and as long as S2, 1.8× as long as wide, S2 twice as long as wide, S3–5 transverse, club elongate oval, regularly pubescent, slightly shorter than funicle.

Thorax. Pronotum black, with two small tubercles between middle and sides, punctures feebly visible, moderately deep, dense, intervals between punctures shining, finely rugose, hardly visible between recumbent, elliptical (l/w 4–7) whitish and dark brown scales, former ones forming small spots more numerous at sides and two tufts covering tubercles; conical, transverse (Pw/Pl 1.67–1.72), widest at base, sides subrectilinear in basal 2/3 then feebly sinuate, moderately convex. Anterior margin of prosternum with deep emargination, without canal. Scutellar shield cordiform, with dense light brown scales.

Elytra. Black, globose (El/Ew 1.08–1.11), widest at middle (Ew/Pw 1.83–1.86), humeri prominent, rounded, sides very slightly rounded, disc distinctly convex; interstriae somewhat convex, with small shallow sparse punctures, intervals smooth, shining; scales recumbent to subrecumbent, elliptical (l/w 4–7), dark brown and whitish, latter ones forming small spots on odd interstriae and covering anterior part of humeri; striae straight, with irregular large deep punctures, wider than interstriae, intervals between punctures narrow, situated lower than interstriae.

Venter. Mesosternal process is slightly convex, slightly emarginate. Mesosternum almost as long as V1. Abdomen with dense, irregularly arranged, moderately deep punctures visible between dense, elongate (more elongate, hair-like along midline and forming dense tuft on V2), whitish scales; V1 1.70–1.75× as long as V2; V1–2 3.45–3.50× as long as V3–4, V3–4 1.35–1.40× as long as V5.

Legs. Femora black with intermixed dark brown and light brown scales, with stout sharp teeth; tibiae black, without uncus; tarsi black with T1 1.4× as long as wide, T2 transverse, T3 distinctly bilobed and distinctly broader than T2, onychium as long as T1–3 combined; claws are asymmetrical, with one claw being 2/3 as long as the other.

Penis. Figure 6d–f, body with slightly convergent sides, blunted tip, long, thin, poorly sclerotized flagellum, and two convergent symmetrical sclerites in its apical half.

Female. Rostrum longer (Rl/Rw 4.62–69; Rl/Pl 1.53–1.59), scales on abdomen only slightly longer in middle than at sides, claws symmetrical.

**Variability.** Length 3.7–4.7 mm. The light scales on the pronotum vary from white to light yellow.

**Biology.** One specimen was collected in Myanmar on *Dolichandrone stipulata* (Wall.) Benth. et Hook. (Bignoniaceae).

**Distribution.** Southern China, Myanmar, Laos, Thailand, Vietnam, Cambodia.

**Non-type material examined.** China: S-Yunnan (Xishuangbanna), 23 km NW Jinghong, Na Ban Village, 22 10.04N 100 39.52E, 680 m, 18.V.2008, leg. Weigel (1, NMEG). Myanmar: Bago, Toungoo (1, BMNH); Mandalay, Yeni Res., Pyinmana, 7.VI.1934, on *Dolichandrone stipulata*, leg. Desai (1, BMNH); Rangoon, VI.1952, leg. Bentoglio (2, NHMB); Sagaing, Alaungdaw Katthapa. 3.–13.V.2003, legg. Boukal and Schillhammer (1, NHMW); Tenasserim (1, BMNH); Tenasserim, Meetan, IV.1887 (1, MRAC); Tenasserim, Papu (1, BMNH); Tenasserim Mountains, Siam border, II-V.1913 (1, BMNH). Laos: Kham Mouan pr. Ban Khoun Ngeun, 19–31.V.2001, leg. Pacholátko (6, NHMB); Louangphrabang pr. Ban Song Cha, 5 km W, 1200 m, 1–16.V.1999, leg. Kubáň (2, NHMB); Louangphrabang pr. 25 km E Muang Ngoy, 1000 m, 23.IV.1999, leg. Kubáň (8, NHMB); Louangnamtha pr. Namtha, Muang Sing, 1200 m, 5–31.V.1997, leg. Kubáň (20, NHMB); 15 km NW Louang Namtha, 13.–24.V.1997, legg. Štrba and Hergovits (1, MKCS); 20 km NW Louang Namtha, legg. Štrba and Hergovits (1, HWCB); Vientiane, V.1915, leg. de Salvaza (1, BMNH). Thailand: Chiang Mai prov., San Pakia, 1400 m, 1.–15.V.1998, leg. Kubáň (1, RCCM); Doi Suthep-Pui, 1300–1500 m, 18.–23.IV. 1991, leg. Pacholátko (4, NHMW); Lansang Nat. Park, 18-24.IV.1991, leg. Král (1, NHMB); Mae Hong Son, 28.IV.–3.V.1992, leg. Pacholátko (1, NHMW); Mae Hong Son, Soppong, 8.V.1997, leg. Becvar (2, FBCV, HWCB); Mae Hong Son, Ban Huai Po, 1600 m, 9.–16.V.1991, leg. Pacholátko (2, NHMB); Mae Hong Son, Ban Huai Po, 1800 m, 30.IV–14.V.1991, leg. Farkač (20, NHMB); Mae Hong Son, Ban Hua Nam, 1200 m, 4–6.V.1991, leg. Farkač (2, NHMB); Mae Hong Son, Ban Si Lang, 1000 m, 1–7.V.1992, leg. Bily (4, NHMB); Palong, 28.V.1991, leg. Kubáň (8, NHMB); Soppong Pai, 1800 m, 25.IV.–5.5.1992, leg. Pacholátko (1, NHMW); Soppong-Pai, 1800 m, 1–6.V.1991, leg. Pacholátko (15, NHMB); Thanon Thong Chai, Chiang Dao, 1000 m, 17–24.V.1991, legg. Kral and Kubáň (5, NHMB); Thimonghta, 9–13.IV.1991, leg. Kubáň (4. NHMB). Vietnam: Bac Kan Pr., Ba Be National Park, 22°24′N 105°37′E, 18.V.2014, leg. Weigel (1, NMEG); Bao-lac (6, MNHN); Bao-lac, Layoye (1, BMNH); Gialai Contum, Buen Luoi, tropical forest, 12.VI.1985, leg. Medvedev (2, ZISP); Buenloi Darevskyi 20.VI.1982 (1, ZISP); Hoabinh, VIII.1918, leg. de Salvaza (1, BMNH); Hoa Binh, VIII.1918, leg. de Salvaza (3, BMNH); Choganh, 1.VIII.1914 (7, MNHN); Ha Nam Ninh, Cuc Phuong, 25.Y.1986, leg. Horák (4, NHMB); mountains 30 km N Khôn-Gái, 300 m, 8.IV.1962, leg. Kabakov (1, ZISP); mountains SW Kui-Chai, 200 m, 12.XII.1963, leg. Kabakov (1, ZISP); Lao-Kay (1, BMNH); Lô River [Rivière Claire] (9, MRAC); Lang Son, Than Moi, IV.1917, leg. de Salvaza (1, BMNH); Lang Son, Than Moi, leg. Perrot (1, MNHN); Ninh-binh, Cuc-Phuong, tropical forest, 24–26.IV.1975, legg. Medvedev and Dang Dap (2, ZISP); Shon-Zuong, 20.III.1962, leg. Kabakov (1, ZISP); Son La, Song Ma, 3.V.1986, leg. Shovrkov (1, ZISP); mountains NW Tam-Dao Shon-Zuong, 300 m, 12 IV 1962, leg. Kabakov (1, ZISP); mountains Tchiem-Khoa, 6.IV.1960, leg. Kabakov (1, ZISP); 40 km NO Thái-Nguyên 300–500 m, 13.IX.1962, leg. Kabakov (4, ZISP); 10 km S Thái-Nguyên, 31.X.1976, leg. Medvedev (1, ZISP); 60 km W Vin-Lin, 100–500 m, 18.III.1963, leg. Kabakov (1, ZISP). Cambodia: Cambodia (2, MNHN); Kompong Kedey (1, MNHN).

**7.** ***Cionus tonkinensis* Wingelmüller** (Figure 7a–f)

*Cionus tonkinensis* Wingelmüller, 1915: 308 [34].

**Type locality.** Bao Lac (Vietnam).

**Type series.** This species was described from specimens of Bedel’s collection (MNHN) collected in Tonkin (Bao-lac, presently a northern district of Vietnam). We examined five syntypes labelled “Bao-lac, Ht. Tonkin/♂/Type/Cionus tonkinensis Wingelm.” (male, already dissected by Wingelmüller; lectotype here designated); “Bao-lac, Ht. Tonkin/♂.” (male, already dissected by Wingelmüller; paralectotype); “Bao-lac, Ht. Tonkin/♂.” (male; paralectotype);4. [without labels] (male, already dissected by Wingelmüller; paralectotype); “♀” (female; paralectotype).

**Diagnosis.** Rostrum stout, short. Pronotum with four small tubercles, numerous light scales on sides, some of them covering tubercles. Elytra with white scales forming many small spots on odd interstriae and largely covering humeri. Claws in males slightly asymmetrical.

**Redescription.** Male.

Body. Stout, globose.

Head. Rostrum stout, short (Rl/Rw 3.20–3.25, Rl/Pl 1.20–1.24), black, in lateral view moderately curved, slightly tapered from antennal insertion to apex; in dorsal view slightly enlarged from base to apex; with recumbent, sparse, subtle (l/w 3–6) whitish and light brown scales near to apex. Head between eyes very narrow, 1/4 as wide as rostrum at base. Eyes flat. Antennae dark brown, inserted between middle and apical 1/3, scape long, funicle shorter than scape, S1 slightly more robust than and as long as S2, 1.8× as long as wide, S2 twice as long as wide, S3–5 transverse, club elongate oval, regularly pubescent, slightly shorter than funicle.

Thorax. Pronotum black, with four small tubercles between middle and sides, with feebly visible, moderately deep dense punctures, intervals between punctures shining, finely rugose, hardly visible between recumbent, elliptical (l/w 4–7), whitish and dark brown scales, white scales more numerous at sides leaving only two areas of dark scales and covering tubercles; conical, transverse (Pw/Pl 1.67–1.71), widest at base, sides in basal 2/3 subrectilinear,then slightly sinuate, moderately convex. Anterior margin of prosternum with deep emargination, without canal. Scutellar shield cordiform, with dense light brown scales.

Elytra. Black, globose (El/Ew 1.08–1.11), widest at middle (Ew/Pw 1.84–1.87), humeri prominent, rounded, sides very slightly rounded, disc area distinctly convex; interstriae moderately convex, with small shallow sparse punctures, intervals smooth, shining; scales elliptical (l/w 4–7), recumbent to subrecumbent, dark brown and whitish, latter ones forming small spots on odd interstriae and almost completely covering humeri; striae straight, with irregular large deep punctures, wider than interstriae, intervals between punctures narrow, situated lower than interstriae.

Venter. Mesosternal process is slightly convex, weakly emarginate. Mesosternum almost as long as V1. Abdomen with dense, irregularly arranged, moderately deep punctures visible between dense, elongate (more elongate, hair-like along midline forming dense tuft on V2) whitish scales. V1 1.60–1.67× as long as V2; V1–2 3.36–3.45× as long as V3–4, V3–4 1.30–1.37× as long as V5.

Legs. Femora black with intermixed dark brown and light brown scales, with stout sharp teeth; tibiae black, without uncus; tarsi black, T1 1.3× as long as wide, T2 transverse, T3 distinctly bilobed and distinctly broader than T2, onychium as long as T1–3 combined; claws are asymmetrical, with one claw being 2/3 as long as the other.

Penis. Figure 7d–f, body with subparallel sides, blunted tip, long, thin poorly sclerotized flagellum bifurcated at base, and two symmetrical poorly sclerotized sclerites with convergent sides in its apical half.

Female. Rostrum longer (Rl/Rw 4.62–4.69; Rl/Pl 1.54–1.60), scales on abdomen only slightly longer at middle than at sides, claws symmetrical.

**Variability.** Length 3.7–4.7 mm. Light scales on the pronotum vary from white to light yellow.

**Comparative notes.** This species is similar to *C. albopunctatus* and *C. obesus*, from which it differs by the stouter rostrum in lateral view being slightly wider at antennal insertion; denser scales at sides of the pronotum, humeri and episterna; four tubercles on the pronotum instead of two; stouter femora and narrower; and more regular striae.

**Distribution.** Myanmar, Thailand, Vietnam.

**Non-type specimens examined.** Myanmar: Tenasserim Mountains, Siam border, II-V.1913 (1, BMNH). Thailand: Mae Hong Son, Ban Huai Po, 1600 m, 9.–16.V.1991, leg. Pacholátko (2, NHMB); Mae Hong Son, Ban Huai Po, 800–1600 m, 9.–15.V.1991, leg. Bily (2, NHMB); Mae Hong Son, Ban Huai Po, 1600–2000 m, 17.–23.V.1991, leg. Dombický (1, NHMB). Vietnam: Na Hang, 160 km NNW Hanoi, 30.–31.V.1996, leg. Napolov and Roma (1, NMEG); Rivière Claire (2, MRAC); Tay Ninh (1, MRAC); 40 km NO Thái-Nguyên 300 m, 14.V.1963, leg. Kabakov (3, ZISP).

**8.** ***Cionus obesus* Pascoe** (Figure 8a–f)

*Cionus obesus* Pascoe, 1883: 93 [35]. Marshall, 1926: 368 [26].

**Type locality.** Chennai (Tamil Nadu, India).

**Type series.** This species was described based on specimens from Madras (currently Chennai). We found only one syntype labelled: “Madras [elliptical blue card]/Type [round white card with red margins]/Cionus obesus, Type Pasc./Pascoe Coll., 93-60 [upturned card]” (MRAC, lectotype here designated).

**Diagnosis.** Pronotum with two small lateral tubercles more distinct due to tufts of erect white scales, and with other small spots of light scales. Elytra with many light spots, interstria 3 moderately carinate at base. Claws slightly asymmetrical in male.

**Redescription.** Male.

Body. Stout, globose.

Head. Rostrum stout, short (Rl/Rw 3.20–3.24, Rl/Pl 1.12–1.15), black, in lateral view moderately curved, slightly tapered from antennal insertion to apex; in dorsal view slightly enlarged from base to apex; with recumbent, sparse, subtle (l/w 3–6) whitish and light brown scales near to apex. Head between eyes very narrow, 1/4 as wide as rostrum at base. Eyes flat. Antennae brown, inserted between middle and apical 1/3, scape long, funicle shorter than scape, S1 slightly more robust than and as long as S2, 1.8× as long as wide, S2 twice as long as wide, S3–5 transverse, club elongate oval, regularly pubescent, slightly shorter than funicle.

Thorax. Pronotum black, punctures feebly visible, moderately deep, dense, intervals between punctures shining, finely rugose, hardly visible between recumbent, elliptical (l/w 4–7) whitish, light brown and dark brown scales, lighter ones forming small spots and two tufts (false tubercles) between midline and sides, more numerous at sides; conical, transverse (Pw/Pl 1.66–1.69), widest at base, sides in basal 2/3 subrectilinear, then slightly sinuate, humeri prominent, rounded, moderately convex. Anterior margin of prosternum with deep emargination, without canal. Scutellar shield cordiform, with dense light brown scales.

Elytra. Black, globose (El/Ew 1.13–1.15), widest at middle (Ew/Pw 1.77–1.80), sides very slightly rounded, disc distinctly convex; interstriae moderately convex, with small, shallow sparse punctures, intervals smooth, shining; scales elliptical (l/w 4–7), recumbent to subrecumbent, dark brown, light brown and whitish, light brown scales forming small spots on odd interstriae and covering anterior part of humeri; striae straight, with irregular, large deep punctures wider than interstriae, intervals between punctures narrow, situates lower than interstriae.

Venter. Mesosternal process is slightly convex, slightly emarginate. Mesosternum almost as long as V1. Abdomen with dense, irregularly arranged, moderately deep punctures feebly visible between moderately dense, elongate (more elongate, hair-like, along midline forming dense tuft on V2) whitish scales; ventrite. V1 1.70–1.73× as long as V2; V1–2 3.42–3.45× as long as V3–4, V3–4 1.33–1.38× as long as V5.

Legs. Femora black, with intermixed dark brown and light brown scales, stout sharp teeth; tibiae black, without uncus; tarsi black, T1 1.3× as long as wide, T2 transverse, T3 distinctly bilobed and distinctly broader than T2, onychium as long as T1–3 combined; claws are asymmetrical, with one claw being 2/3 as long as the other.

Penis. Figure 8d–f, body with subparallel sides, slightly emarginated blunted tip, long, thin, poorly sclerotized flagellum bifurcated at base, and two symmetrical, poorly sclerotized sclerites with convergent sides in its apical half.

Female. Rostrum longer (Rl/Rw 3.92–3.94, Rl/Pl 1.14–1.17), V2 without tuft of scales, claws symmetrical.

**Variability.** Length 4.4–4.5 mm. There are no relevant differences between the examined specimens.

**Comparative notes.** This species differs from the closely related *C. albopunctatus* by less conical pronotum with slightly rounded sides and more convex disc.

**Distribution.** Southeastern India.

**Non-type material examined.** India: India orient., Fry Coll. (2, BMNH); Victoria Pt., Atkinson coll. (2, BMNH).

### 3.2. Key to the Species

1.Elytra with large spot of black scales on dorsum (Figure 1a) ……………………………………………………………*C. indicus* Desbrochers des Loges-Elytra without large spot of black scales on dorsum ……………………………...……22.Rostrum long and thin, especially in female (Figure 2b,c and Figure 3b)……..……………...3-Rostrum short and stout in both sexes …………………….………………….………….43.Pronotum between midline and sides with two small spots of recumbent white scales. Even elytral interstriae covered with brown scales; interstria 1 at midlength with two rectangular spots of white scales (Figure 3a)…......…………….……3. *C*. *vossi* **nom. nov.**-Pronotum between midline and sides with two little spots of suberect white scales simulating small tubercle. Even elytral interstriae covered with whitish grey scales; interstria 1 at midlength without rectangular spots of white scales (Figure 2a)……………………….………………………………….……...2. *C*. *radermacherae* Voss4.Pronotum without tubercles (Figure 4a and Figure 5a) ………………………….……………...5-Pronotum with two or four small tubercles (Figure 6a, Figure 7a and Figure 8a)……………………..……...………………..……………………..……………………....65.Integument black. Rostrum shorter, not tapered from antennal insertion to apex (Figure 4b,c). Pronotum with small spots of light scales on disc and at sides. Elytral interstriae 8 and 9 in basal third only slightly sinuate (Figure 4a,b)……………………………………………………….........................4. *C*. *meleagris* Marshall-Integument brown. Rostrum moderately longer; in lateral view, it is tapered from antennal insertion to apex (Figure 5b,c). Pronotum at sides densely covered with light scales. Elytral interstriae 8 and 9 in basal third distinctly sinuate (Figure 5a,b)……………………………………..…………………………5. *C*. *ottomerkli*
**sp. nov.**6.Rostrum stouter, in lateral view slightly wider at antennal insertion (Figure 7b,c). Light scales denser at sides of pronotum, humeri and episterna. Pronotum with four small tubercles between midline and sides emphasized by tufts of raised white scales. Striae narrower and more regular (Figure 7a,b). Femora stouter …………………………………..……………..……………..7. *C*. *tonkinensis* Wingelmüller-Rostrum stouter, in lateral view equal in width. Light scales sparse at sides of pronotum, humeri and episterna. Pronotum with two small tubercles between midline and sides emphasized by tufts of raised white scales. Striae wider and irregular. Femora thinner……………………………………………………………………………………….77.Pronotum less conical, with slightly rounded sides and more convex disc (Figure 8a)……………….…………………………….…………………………...8. *C*. *obesus* Pascoe-Pronotum distinctly conical, with rectilinear sides and less convex disc (Figure 6a)….…………………………………………………..….…6. *C*. *albopunctatus* Aurivillius

## 4. Discussion

The genus *Cionus* is poorly represented in the Oriental region in comparison to the numerous species occurring in the Palaearctic and the Afrotropical regions [1,2]. It is nevertheless possible that targeted collections could uncover other species; however, it should be pointed out that in all important collections we examined, specimens from the Oriental realm were rare. It seems that *Cleopus* and *Stereonychus*, represented with a few species in the Palaearctic region and lacking in the Afrotropical region, seem to be represented in the Oriental region by considerably more species than *Cionus*.

Among eight species presently known from this region, one (*C. indicus*) appears at first sight to be related to Palaearctic species due to the presence of a large broad spot of dark scales on the elytral disc. However, this spot may only be analogous and not homologous with the spot present in Palaearctic species. This supposition is supported by the fact that first and second elytral striae are almost rectilinear and not clearly sinuate in correspondence of the lateral margins of the spot. On the contrary, with the other seven species being mutually similar in regard to the pattern of the vestiture, they seem to be more related to Afrotropical species. It is noteworthy that in three species (*C. albopunctatus*, *C. tonkinensis* and *C. obesus*), the pronotum bears two or four small tubercles which are found in several species from South Africa and Madagascar but are absent in all Palaearctic species. However, the male genitalia (body of penis with unusual apical sclerites) are clearly more similar to other Oriental species (see *C. radermacherae* and *C. meleagris*) than to Palaearctic species.

Information on the relationships between the species of various regions could be obtained by the knowledge of the host plants. It is well-known that most species of *Cionus* feed on species of the family Scrophulariaceae—Scrophularieae (mainly *Verbascum* and *Scrophularia*) for the Palaearctic species, and Buddlejeae (*Buddleja*) for the Afrotropical species—whereas one Eastern Palaearctic species (*C. helleri* Reitter, 1904) lives on Paulowniaceae (*Paulownia tomentosa* (Thunb.) Steud.) and two Afrotropical species (*C. perlatus* Faust, 1885; and *C. tristis* Boheman, 1838) live on Bignoniaceae (*Stereospermum kunthianum* Cham. and *Rhigozum obovatum* Burch respecrively) [1,2]. Unfortunately, the available biological data on the Oriental species are very little. However, two of them, *C. radermacherae* and *C. albopunctatus*, were collected on Bignoniaceae—*Radermachera gigantea* (Blume) Miq. [29] and *Dolichandrone stipulata* (Wall.) Benth. et Hook., respectively—whereas a third species, *C. indicus*, was collected on Lamiaceae (*Premna pyramidata* Wall. ex Schauer). Although this last datum needs confirmation, it is noteworthy that all of these host families belong to Lamiales and that the two abovementioned Afrotropical species living on Bignoniaceae are particularly similar in their morphology to the Oriental species.

We think that, only after a careful phylogenetic molecular study, it will be possible to hypothesize the origin of the dispersal of this genus. On the other hand, the presence of pronotal tubercles in some Eastern Palaearctic and Oriental species is also shared by members of the related genera *Stereonychus* and *Cleopus*, of which some still are to be described, and may shed further light on the relationship of the genera of the tribe.

## Figures and Tables

**Figure 1 insects-14-00646-f001:**
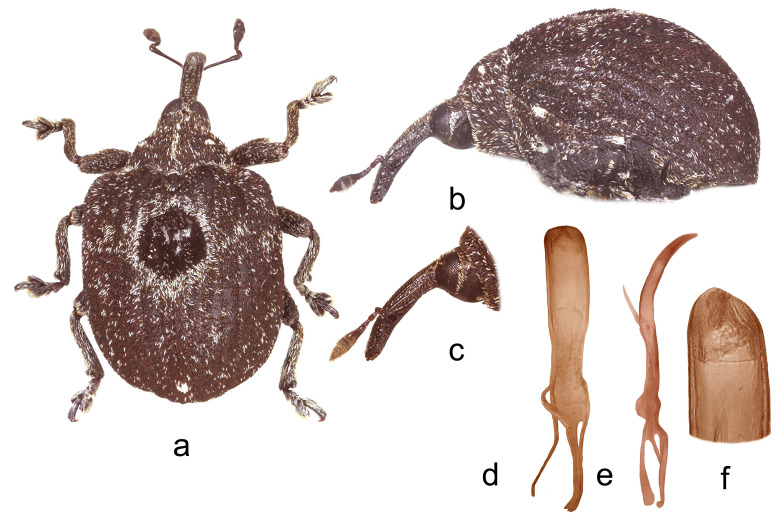
*Cionus indicus*: (**a**) body in dorsal view (male), (**b**) body in lateral view (male), (**c**) rostrum in lateral view (female), (**d**) penis in ventral view, (**e**) penis in lateral view and (**f**) apex of penis in dorsal view. Not to scale.

**Figure 2 insects-14-00646-f002:**
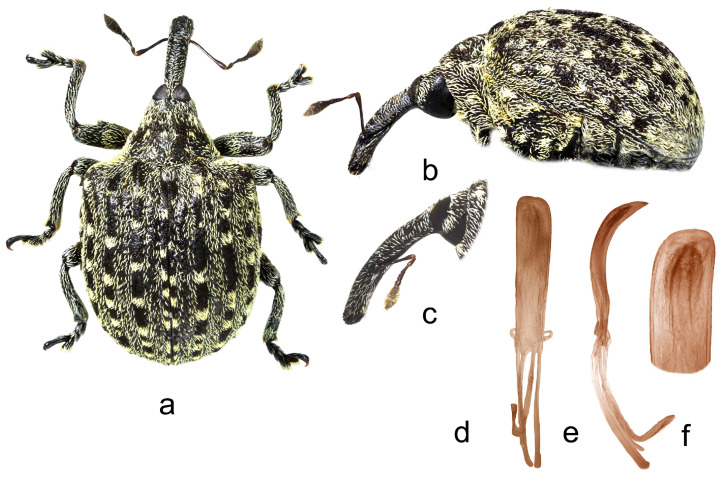
*Cionus radermacherae*: (**a**) body in dorsal view (male), (**b**) body in lateral view (male), (**c**) rostrum in lateral view (female), (**d**) penis in ventral view, (**e**) penis in lateral view and (**f**) apex of penis in dorsal view. Not to scale.

**Figure 3 insects-14-00646-f003:**
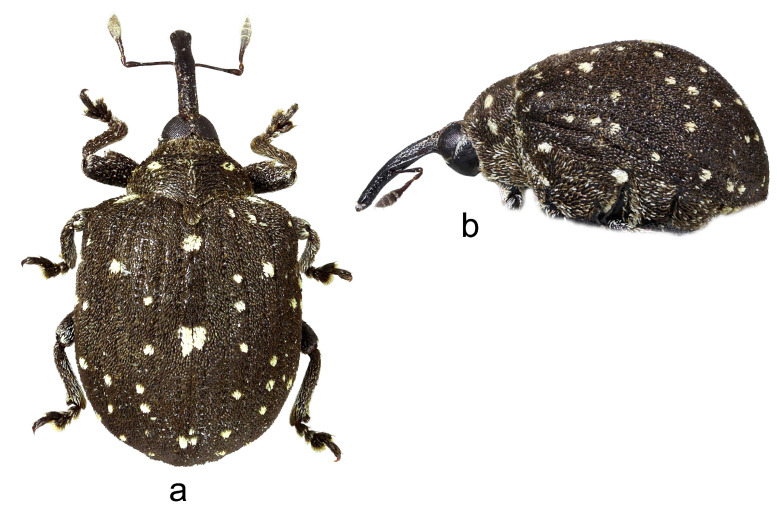
*Cionus vossi* **nom. n.**: (**a**) body in dorsal view (female) and (**b**) body in lateral view (female). Not to scale.

**Figure 4 insects-14-00646-f004:**
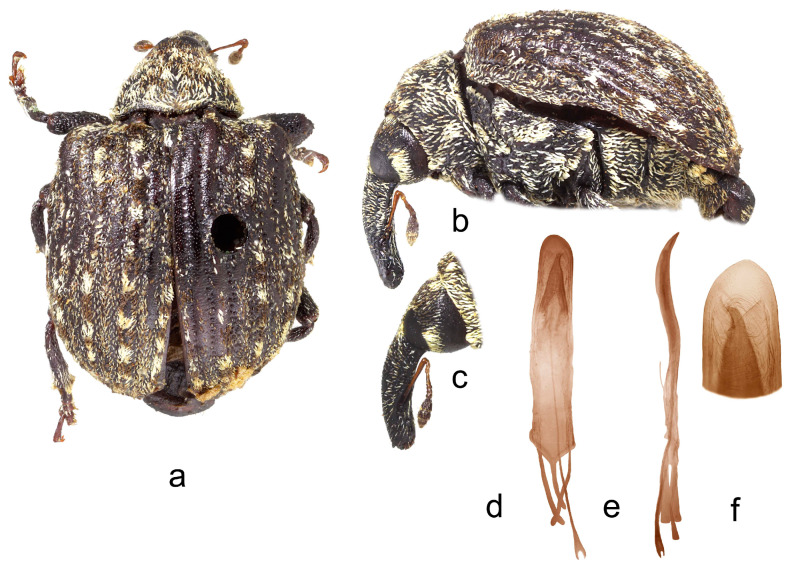
*Cionus meleagris*: (**a**) body in dorsal view (male), (**b**) body in lateral view (male), (**c**) rostrum in lateral view (female), (**d**) penis in ventral view, (**e**) penis in lateral view and (**f**) apex of penis in dorsal view. Not to scale.

**Figure 5 insects-14-00646-f005:**
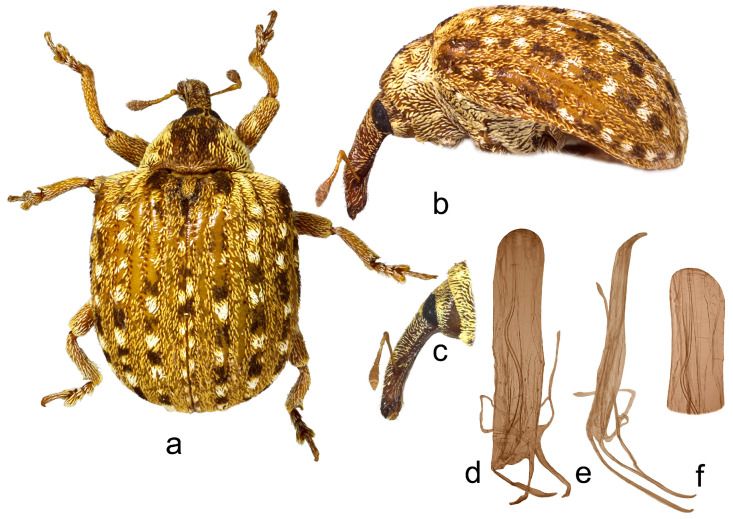
*Cionus ottomerkli*: (**a**) body in dorsal view (male), (**b**) body in lateral view (male), (**c**) rostrum in lateral view (female), (**d**) penis in ventral view, (**e**) penis in lateral view, and (**f**) apex of penis in dorsal view. Not to scale.

**Figure 6 insects-14-00646-f006:**
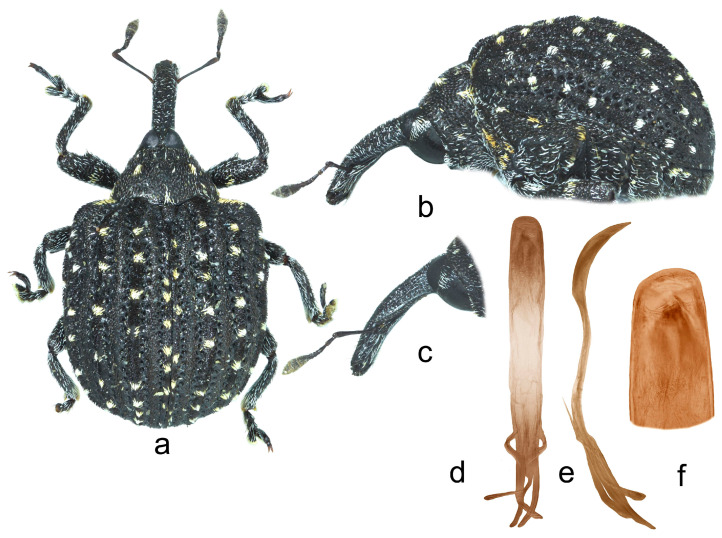
*Cionus albopunctatus*: (**a**) body in dorsal view (male), (**b**) body in lateral view (male), (**c**) rostrum in lateral view (female), (**d**) penis in ventral view, (**e**) penis in lateral view and (**f**) apex of penis in dorsal view. Not to scale.

**Figure 7 insects-14-00646-f007:**
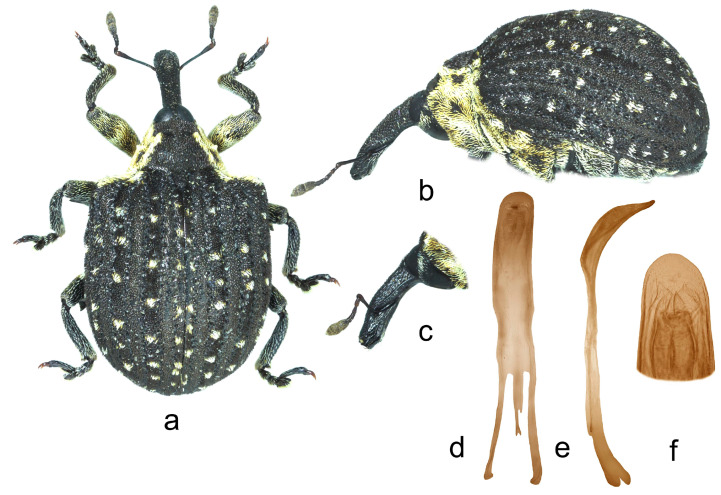
*Cionus tonkinensis*: (**a**) body in dorsal view (male), (**b**) body in lateral view (male), (**c**) rostrum in lateral view (female), (**d**) penis in ventral view, (**e**) penis in lateral view and (**f**) apex of penis in dorsal view. Not to scale.

**Figure 8 insects-14-00646-f008:**
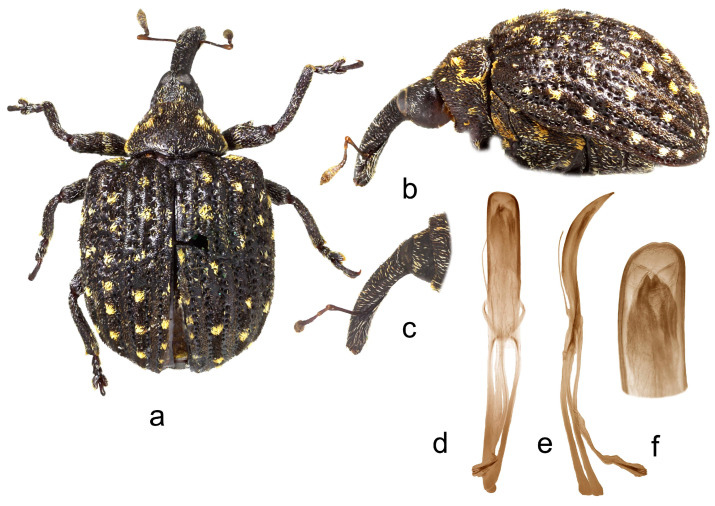
*Cionus obesus*: (**a**) body in dorsal view (male), (**b**) body in lateral view (male), (**c**) rostrum in lateral view (female), (**d**) penis in ventral view, (**e**) penis in lateral view and (**f**) apex of penis in dorsal view. Not to scale.

## Data Availability

All data used in this study are based on dried insect specimens deposited in publicly accessible institutional depositories (listed in the section Depositories) or in depositories of our colleagues (ibid). All data used in this study are not a subject of any legal or commercial restriction.

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
