# Peer review of "A Taxonomic Revision of the Genus Cionus (Coleoptera, Curculionidae) from the Oriental Region"

_insects, 2023, doi:10.3390/insects14070646_

Round 1

Reviewer 1 Report

The work is a relatively well presented taxonomic study of a weevil genus.  The descriptions and key are well done and figures are adequate. This reviewer considered that there was not enough discussion of the hosts plants of the genus, or at least acknowleging that there is little data. This reveiwer also considered that the English could be improved, although this may be due, in part, to local differences in word use.  Please consider my suggestions in the attached file.

See attached file. 

Reviewer 2 Report

Very good and useful work.

Individual comments below.

Line 154 - most species of Cionus have unequally long claws in males, especially on protarsi - For me, the claw length in Cionini is quite "normal" compared to the other weevils; however distinctly elongated is the last tarsal segment/distal tarsomere/onychium

Line 196, 269, 344, 413, 471, 534, 636, 704 - Male. Body. Stout, globose. - should be: Male. Body stout, globose. (???)

Line 805 - C. perlatus without full name.

Line 810 - C. meleagris - as above

Detailed editorial comments:

The font size should be uniform throughout the manuscript, except for selected sections, in accordance with the journal's standards. Occasionally enlarged/reduced font in individual lines (e.g. 63-64); some parts of the  manuscript have different line spacing (e.g. from line 299 to 384)

Line 76 - is "ofBhutan", should be "of Bhutan".

Line 115 - lack of "tab", wrong text alignment.

Line 127 - is "Cionus", should be "Cionus".

Line 142 - is "Cioninioccuring", should be "Cionini occuring".

Line 415, 423, 426 - incorrect splitting words between lines: moder-ately - splitting should be between syllables: mo-de-ra-te-ly; 726 - si-tu-a-tes.

The problems mentioned above are also repeated in other parts of the manuscript
